

# SDN-IoT: SDN-based efficient clustering scheme for IoT using improved Sailfish optimization algorithm

Ramin Mohammadi[1], Sedat Akleylek[2,3,4] and Ali Ghaffari[5,6]

[1] Ondokuz Mayis University, Department of Computational Sciences, Samsun, Türkiye
[2] Department of Computer Engineering, Ondokuz Mayis University Samsun, Samsun, Türkiye
[3] University of Tartu, Tartu, Estonia
[4] Ondokuz Mayis University, Cyber Security and Information Technologies Reseach and Development Center, Samsun, Türkiye
[5] Department of Computer Engineering, Tabriz branch, Islamic Azad University, Tabriz, Iran
[6] Department of Computer Engineering, Faculty of Engineering and Natural Sciences, Istinye University, Istanbul, Türkiye

Corresponding author
Sedat Akleylek, akleylek@gmail.com

## ABSTRACT

The Internet of Things (IoT) includes billions of different devices and various applications that generate a huge amount of data. Due to inherent resource limitations, reliable and robust data transmission for a huge number of heterogenous devices is one of the most critical issues for IoT. Therefore, cluster-based data transmission is appropriate for IoT applications as it promotes network lifetime and scalability. On the other hand, Software Defined Network (SDN) architecture improves flexibility and makes the IoT respond appropriately to the heterogeneity. This article proposes an SDN-based efficient clustering scheme for IoT using the Improved Sailfish optimization (ISFO) algorithm. In the proposed model, clustering of IoT devices is performed using the ISFO model and the model is installed on the SDN controller to manage the Cluster Head (CH) nodes of IoT devices. The performance evaluation of the proposed model was performed based on two scenarios with 150 and 300 nodes. The results show that for 150 nodes ISFO model in comparison with LEACH, LEACH-E reduced energy consumption by about 21.42% and 17.28%. For 300 ISFO nodes compared to LEACH, LEACH-E reduced energy consumption by about 37.84% and 27.23%.

## INTRODUCTION

The Internet of Things (IoT) contains billions of various devices that are connected to the Internet and enable machine-to-machine (M2M) communication *via* data gathering and management (*Hosseinzadeh, Hemmati & Rahmani, 2022*). The IoT uses various technologies and devices, such as radio frequency identification (RFID), tiny sensors, and various wireless communications (*Azari & Ghaffari, 2015*; *Jazebi & Ghaffari, 2020*; *Nikokheslat & Ghaffari, 2017*). The IoT comprises three layers: the perception layer, the network layer for communication, and the application layer. The perceptual layer consists of devices that collect environmental data using sensors and actuators that cause physical

changes in the environment. Between the perception layer and the application layer, the communication network layer is in charge of the network services. Application layers make smart decisions based on low-layer data. IoT sensors are deployed across a wide geographical area to sense, collect, and transmit data. The complexity of sensor nodes is increasing, and it is impossible to manually control and maintain such dense nodes in real-time mode (*Balaji, Nathani & Santhakumar, 2019*). The design of IoT applications should consider the capabilities of various resources, including various devices that form part of the IoT environment. Scalability is a crucial issues in the IoT, as providing a communication model for billions of heterogenous devices is a critical mission (*Chowdhary & Rao, 2021*).

Limited-power IoT devices are connected to various things such as smart cameras, intelligent vehicle, and intelligent homes (*Almutairi & Aldossary, 2021*). Also, the daily growth of various IoT services, along with the increase in traffic produced by IoT devices, makes issues about increased energy consumption, distribution, and placement of smart objects. Wireless communications and computing resources are usually very limited and low energy. Therefore, it is difficult to prepare the rising request of IoT services and the heterogeneous needs of smart objects (*Bajaj, Sharma & Singh, 2021*). One possible solution for the IoT is to use SDN. Smart network control by SDN ensures consistent and fair operation. SDN technology facilitates IoT network by separating the control level from the data surface, easier programming, and centralized network architecture (*Shirmarz & Ghaffari, 2022*). The SDN controller is able to manage the network with software commands.

In SDN, date plane is separated from control plane for easing network management (*Jafarian et al., 2021*). The task of the control plane is to handle network functions such as input stream management, network analysis, and synchronization. The control plane operates as the monitoring center of the network with intelligent programming (*Jazaeri et al., 2021*). SDN controllers with intelligent programming provide a better chance of defining different decisions according to network conditions.

SDN architecture with intelligent programming and intelligent network management is a key factor in the dynamic and complex nature of the IoT (*Ahmadian & Ahmadi, 2022*). The SDN-IoT architecture includes three main layers: The first layer is the device layer, which at the lowest level of the SDN-IoT architecture consists of IoT devices (*Tang et al., 2018*; *Wu et al., 2021*). They sense and collect great amounts of data that may be in different formats, and take the collected data to higher levels for further analysis and processing. The control plane decides what to do with the collected data and sends the relevant instructions and submission rules through the Southern APIs to the routers and gateways of the network. These routers and gateways operate according to the OpenFlow protocol and send data according to instructions received from the control level. The second layer is the control/middleware plane, and SDN controllers are the basic components of this layer, simplifying the complexity of network management (*Ren et al., 2019*). In general, the task of the control plane is organizing various services in the IoT, managing network topology and routing, and managing network traffic. The third layer is the application layer and

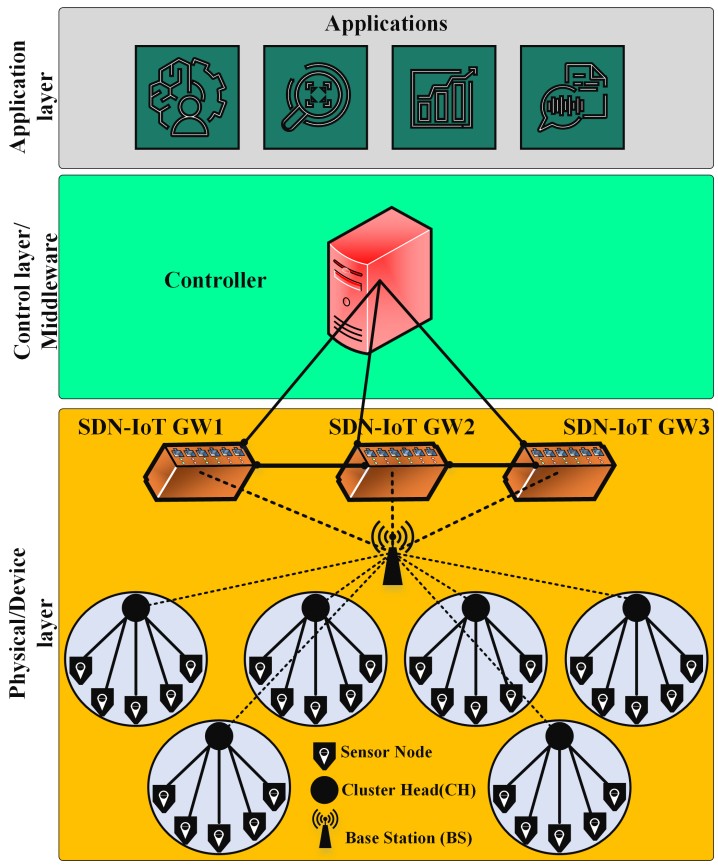

**Figure 1  An overview of the SDN-IoT architecture.**

includes various services and applications to serve users. Figure 1 shows an overview of the SDN-IoT architecture.

A variety of SDN-based clustering protocols have been proposed (*Al-Janabi & Al-Raweshidy, 2017*; *Ouhab et al., 2020*). In these models, the SDN controller is used to divide the network into different regions to balance the number of CH nodes in each area. An important challenge in clustering is choosing the optimal CH node that directly affects the performance of the IoT network. Due to data aggregation and transmission, CH nodes consume more energy than cluster members. Hence, the CH nodes must be periodically changed by an intelligent algorithm and selected effectively to balance the overall network energy.

In this article, SDN-based architecture and meta-heuristic algorithm for optimal clustering are used to intelligently manage IoT. The Sailfish algorithm (*Shadravan, Naji & Bardsiri, 2019*), is an efficient meta-heuristic algorithm for solving different optimization problems (*Geetha, Nanda & Yadav, 2022*; *Reza Naji et al., 2022*). The SDN controller uses optimization algorithms to make optimal decision-making capabilities and helps to collect data and reduce energy consumption by creating an optimal clustering technique. This article uses ISFO and presents a clustering scheme for IoT sensor nodes in SDN-IoT

environment to reduce power consumption. The SDN controller uses an improved Sailfish algorithm to select CH nodes. In the SDN-IoT architecture, the sensor nodes are programmatically controlled by the SDN controller and reconfigure their performance based on the actual measurement of the environment. The main contributions of this article are as follows:

- Use SDN to manage IoT nodes called SDN-IoT
- Clustering method using ISFO algorithm
- Reduce power consumption and increase the life of IoT nodes using SDN
- CH selection process based on various criteria such as residual energy, distance within the cluster and distance from the CH to base station (BS)

The rest of the article is organized as follows: Section 2, explains related works and SFO algorithm. Section 3, describes the proposed method in details. Section 4, indicates the performance evaluation of the proposed method. Finally, Section 5, concludes the article and offers future works.

## LITERATURE REVIEW

In this section, we will study the related works and SFO algorithm.

## RELATED WORKS

In *Sellami et al. (2022)*, the authors investigated the problem of job scheduling based on energy awareness and low latency in an SDN-Fog-IoT network. First, the task of assigning and scheduling tasks online is formulated as an energy-limited Q-Learning process. Second, the goal is to minimize the average end-to-end delay. The task achievement process is then defined by the Deep Reinforcement Learning (DRL) approach for scheduling and assigning dynamic tasks in the Fog network with SDN. The SDN agent collects environment state information during runtime and relays it to the controller. By attempting various activities, the agent learns to maximize the environment's reward. The controller may choose to accept the present policy as the optimal choice for placing tasks on the selected fog-enabled nodes, or it may continue learning from the available dispersed nodes to identify a more suitable option for placing the current task requests. All agents are motivated by the same objective, which is to maximize the anticipated discounted return. Extensive simulation results showed that the new solution performed better than other algorithms. In addition, the energy awareness feature in the new model has been improved, as up to 87% energy savings have been achieved compared to other approaches. The scheduling scheme has been able to perform more tasks with less time delay. The computational time of the algorithm is high and leads to an increase in energy.

In *Shi, Zhu & Wei (2022)*, a conscious path selection algorithm called SARSA-aware Delay-Aware (SDRS) using SDN is proposed to simplify IoT network configuration and management. SDRS selects optimal path based on network status information. In addition, SDRS creates a Q-table to control the data transmission path. The SDRS model is used to learn the optimal route selection strategy, which reduces latency and reliability. The results showed that the SDRS model had better performance in terms of transmission delay

and reliability compared to the shortest path selection algorithm (SRS) and random path selection algorithm (RRS). Because SDRS is able to learn the route selection strategy by analyzing the feedback delay performance, it is able to adjust the path, switch the channel type, and fast converge to the best route selection possible.

Edge Computing (EC) is an ecosystem for overcoming barriers to cloud technology to support real-time IoT systems. However, the EC faces issues such as allocating resource for various applications at the edge of the network consisting of resource-constrained nodes. In *Maia et al. (2021)*, a multi-objective genetic algorithm with an initial population based on stochastic and heuristic solutions to obtain near-optimal solutions has been proposed. The evaluation results showed that the new model was superior to other algorithms in terms of operating costs and availability of services. The proposed approach does not take into account the reaction time deadline requirement, which is especially vital for applications that are sensitive to latency.

A new energy efficient dual-purpose discharge strategy based on the Firefly Algorithm (FA) has been proposed (*Adhikari & Gianey, 2019*). The main goal of the new algorithm is to find an appropriate computing device for each IoT application using the two criteria of time complexity and consuming energy. A fitness function is designed based on dual-purpose optimization parameters including computational time of implementing devices. An improved attractiveness function is defined to estimate the attractiveness of agents instead of distance-based attractiveness, which aims determine the optimal position of a computing device. The new strategy finds a suitable computational server based on FA for each task and assigns the task to that server.

Reducing the response time is one of the critical issues for SDNs to balance the load on the controllers. In this regard, switch migration is an efficient way to solve this problem. Selecting an improper target controller and a large number of switch migrations between controllers reduce throughput by increasing the average network response time. In *Kabiri, Barekatain & Avokh (2022)*, using the intelligent combination of GA and PSO, the best controller with suitable capacity for migration is selected. The genetic algorithm generates a number of vectors that represent the controllers, which are estimated based on the controller loads. After that, the best vector that represents the controller and the keys connected to it for migrating the switch with the appropriate capacity is found using the cost function evaluation. Therefore, the best solution is found to transfer the switches to the optimal controllers. The results showed that the floodlight controller decreased in operational capacity including 24.72% improvement and the number of migrations by 13.96%. The most important limitation of this method is that it is not suitable for large environments.

In *Sixu, Muqing & Min (2022)*, the PSO and ABC algorithms are designed for mobile-based SDWSN clustering. SDWSN turns the network into a control page and a data page. There are sensor nodes on the data page that only need to transmit their data to the controller. The controller processes the routing estimation. Sensor nodes can easily receive routing to the sink from the controller. The PSO-based clustering algorithm is used to select the CH nodes. The ABC-based navigation algorithm is used to design the routing of packets to the sink. The new protocol includes benefits such as reduced power

consumption, increased grid life and reduced overhead. This methodology decreases the amount of energy that is consumed by the sensor nodes and increases the lifetime of the network. Sensor nodes have a lower energy requirement for the initial configuration of the routing, and the controller is in charge of planning the routing for the entire network. During the period of setting up the routing, this method cuts down on both the control overhead and the computation overhead. The suggested protocol additionally makes use of an on-demand clustering, which, when taken into consideration as a whole, takes into account the standard deviation of the residual energy as well as the threshold value of the round.

A hierarchical routing scheme based on SDN technology for WSN based on Lion Optimization Algorithm (LOA) was presented (*Srinivasa Ragavan & Ramasamy, 2020*). The main purpose of this work is to minimize energy consumption and thus increase the network lifetime. So, the algorithm is divided into three stages, which are cluster creation, path creation and data transfer. LOA algorithm is used for clustering process. This idea saves energy and reduces routing overhead. Through the implementation of the suggested approach's SDN-based routing optimization, the Quality of Services (QoS) of the network is improved. The hierarchical routing strategy that has been suggested provides for increased control as well as scalability across the network.

The ESRA algorithm is an energy-efficient routing algorithm for IoT applications in SDN based WSNS, especially for the monitoring environment. ESRA algorithm effectively selects network headers to solve the controller placement problem with the aim of achieving network reliability and increasing network lifetime (*Samarji & Salamah, 2021*). The choice of controller among the CH is formulated as an NP-hard problem by considering the remaining energy of the CH nodes, their distance to the sink, and their load. The clustering scheme GA has been adopted to improve network lifetime, end-to-end delay, and PDR.

An optimal CH node selection scheme based on the combination of gray wolf and crow search algorithm (HGWCSOA-OCHS) has been proposed to increase the network lifetime by focusing on minimizing latency, reducing the distance between nodes and residual energy (*Subramanian et al., 2020*). The GWO algorithm is combined with the CSA algorithm to solve the problem of early convergence to explore the search space efficiently. The new model uses the GWO and CSA algorithms in the CH selection process, the balance between exploitation rate and exploration in the search space. The HGWCSOA-OCHS scheme has been compared with Firefly Optimization (FFO) schemes, Artificial Bee Clone Optimization (ABCO), Gray Wolf Optimization (GWO), Gray Wolf-Firefly optimization schemes. The HGWCSOA-OCHS scheme for reducing energy consumption has confirmed the improvement of grid life by balancing the percentage of alive and dead sensor nodes in the grid. A PSO-based Tabu Search (TS) algorithm has been proposed to optimize routing and CH selection (*Vijayalakshmi & Anandan, 2019*). The TS-PSO model finds the optimal path and increases the network lifetime. The most important limitation of the model is the increase in computing time.

In *Liu et al. (2021)*, a clustering-based flow control approach has been proposed by SDN. This approach divides the network into clusters with a minimum number of boundary nodes. Instead of managing the separate streams of each node, the SDN controller manages

only the traffic streams of the clusters through the boundary nodes. Routing algorithm manages the data streams within each cluster. The results of simulation show that the SDN approach is efficient and reduces control messages and managed nodes. This scheme reduces cost of communication for configuring an SDN-WSN by a minimum of 27% and a maximum of 88%, respectively, without reducing packet latency and delivery rates. The SDN controller generates scalable communication overhead, which can be tailored by adjusting cluster size.

## Sailfish algorithm

The SFO algorithm has advantages such as a balance between exploitation and exploration and the avoidance of local optimization. In the SFO, the sailfish represent the candidate solutions. This algorithm has two kinds of search agents called sailfish and sardine. The initial population in the solution space is generated randomly. In a next $d$ search space, the $i$th member in the $k$th round contains the current position $SF_{i,k} \in R(i = 1, 2, \ldots, m)$. The $SF$ matrix is designed to maintain the position of all the sailfish.

$$SF_{position} = \begin{bmatrix} SF_{1,1} & SF_{1,2} & \cdots & SF_{1,d} \\ SF_{2,1} & SF_{2,2} & \cdots & SF_{2,d} \\ \vdots & \vdots & \vdots & \vdots \\ SF_{m,1} & SF_{m,2} & \cdots & SF_{m,d} \end{bmatrix} \tag{1}$$

In Eq. (1), m, d and $SF_{i,j}$ represent the number of sailfish, variables, and the next $j$ value of the $i$ sail. Equation (2) represents the fitness function of each sail as follows:

$$\text{Fitness value of sailfish} = f(\text{sailfish}) = f(SF_1, SF_2, \ldots, SF_m) \tag{2}$$

To evaluate each sailfish, a matrix is defined for the suitability of all solutions as follows:

$$SF_{fitness} = \begin{bmatrix} f(SF_{1,1}, SF_{1,2}, \ldots, SF_{1,d}) \\ f(SF_{2,1}, SF_{2,2}, \ldots, SF_{2,d}) \\ \vdots \quad \vdots \quad \vdots \quad \vdots \\ f(SF_{m,1}, SF_{m,2}, \ldots, SF_{m,d}) \end{bmatrix} = \begin{bmatrix} F_{SF_1} \\ F_{SF_2} \\ \vdots \\ F_{SF_m} \end{bmatrix} \tag{3}$$

In Eq. (3), $m$ saves the number of sailfish, $SF_{i,j}$ after $j$th of the sail $i$th, f stores the fitness function and $SF_{fitness}$ stores the fitness value of each factor based on the target function. The first row of the $SF_{position}$ matrix is passed to the fitness function, and the output shows the fitness value of each factor in the $SF_{fitness}$.

The sardine consensus is another important component of the SFO algorithm and sardines also swim in the search space. Therefore, the location and suitability value of each sardine are defined according to Eq. (4).

$$S_{position} = \begin{bmatrix} S_{1,1} & S_{1,2} & \cdots & S_{1,d} \\ S_{2,1} & S_{2,2} & \cdots & S_{2,d} \\ \vdots & \vdots & \vdots & \vdots \\ S_{n,1} & S_{n,2} & \cdots & S_{n,d} \end{bmatrix} \tag{4}$$

where n, and $S_{i,j}$ and $S_{position}$ are the number of sardines, the $j$th dimension of sardine $i$, the position of all sardines in the matrix, respectively. The fitness value of each sardine is defined according to Eq. (5).

$$S_{fitness} = \begin{bmatrix} f(S_{1,1}, S_{1,2}, \ldots, S_{1,d}) \\ f(S_{2,1}, S_{2,2}, \ldots, S_{2,d}) \\ \vdots \quad \vdots \quad \vdots \quad \vdots \\ f(S_{n,1}, S_{n,2}, \ldots, S_{n,d}) \end{bmatrix} = \begin{bmatrix} F_{S_1} \\ F_{S_2} \\ \vdots \\ F_{S_m} \end{bmatrix} \tag{5}$$

where $S_{i,j}$ represents the $j$th dimension, f is the target function, and $S_{fitness}$ stores the fitness value of each sardine. In the SFO, the new position of the sailfish ( $X^i_{new_{SF}}$) is updated according to Eq. (6). In this phase, search agents provide the exploration phase, which involves searching a large part of the search space for promising solutions that have not yet been updated. Sailfish does not attack only from top to bottom or from right to left and vice versa. They can attack in all directions and in a shrinking circle. As a result, the sailfish updates its position in a circle around the best solution.

$$X^i_{new_{SF}} = X^i_{elite_{SF}} - \lambda_i \times \left( rand(1,0) \times \left( \frac{X^i_{elite_{SF}} - X^i_{injured_S}}{2} \right) - X^i_{old_{SF}} \right) \tag{6}$$

In Eq. (6), $X^i_{elite_{SF}}$ elite sailfish position, $X^i_{injured_S}$ best sardine position, $X^i_{old_{SF}}$ sail current position, $rand(1,0)$ is a random number, and $\lambda_i$ is a factor in the $i$th iteration that is produced by Eq. (7).

$$\lambda_i = 2 \times rand(0,1) \times PD - PD \tag{7}$$

Fluctuation of $\lambda$ and the update position of the sailfish can lead to the divergence of the sailfish and their convergence around the prey. This method leads to discovery and search for solutions at the global level. $PD$ indicates the number of baits per repetition. The adaptive formula for the $PD$ parameter is defined according to Eq. (8). Since the number of prey is reduced during group hunting by sailfish, the PD parameter is an important parameter to update the position of the sailfish around the prey.

$$PD = 1 - \left( \frac{N_{SF}}{N_{SF} + N_S} \right) \tag{8}$$

where $N_{SF}$ and $N_S$ are the number of Sailfish and sardines per phase of the algorithm, respectively. In the SFO algorithm, the new position of sardine $X^i_{new_S}$ is defined according to Eq. (9).

$$X^i_{new_S} = r \times \left( X^i_{elite_{SF}} - X^i_{old_S} + AP \right) \tag{9}$$

where $X^i_{elite_{SF}}$ is the best position for an elite sail, $X^i_{old_S}$ is the current position for sardines, r is a random number, and Attack Power (AP) is the rate of attack of a sail's fish per Represents the iteration defined by Eq. (10). The sardines update rule depends on the $AP$ of the sail.

$$AP = A \times (1 - (2 \times Itr \times \varepsilon)) \tag{10}$$

where A and $\varepsilon$ are coefficients to reduce the AP value linearly from A to 0. Using the $AP$, the number of sardines updating $\alpha$ and $\beta$ are defined as follows:

$$\alpha = N_s \times AP \qquad (11)$$

$$\beta = d_i \times AP \qquad (12)$$

where $d_i$ and NS are the number of variables and the number of sardines, respectively. In the SFO, bait hunting is assumed to occur when sardines become more suitable than sailfish. In this case, the position of the sailfish replaces the last location of the sardine being hunted to increase the chances of catching new prey (Eq. 13):

$$X_{SF}^i = X_s^i \, iff \, (S_i) < f(SF_i) \qquad (13)$$

where $X_s^i$ indicates the current location of the sardine in the i iteration and $X_{SF}^i$ indicates the current location of the sailfish in the $i$ iteration.

## Proposed model

This article proposes an SDN-IoT-based framework for ISFO clustering of IoT sensor nodes. In the ISFO model, a set of sensor nodes $SN = s_1, s_2, \ldots, s_n$ are located in an area of size $L \times L$, where L is the size of the area. The purpose of sensor nodes is to collect IoT devices data. In the proposed model, the role of the SDN network is as a designer. SDN collects various network information and automatically sets up the network. The proposed model is installed inside the SDN controller and the network clustering operation is performed by SDN. The SDN controller is responsible for programming the sink, and the sink sends the appropriate message to the IoT nodes. The SDN controller creates the network topology. The clustering operations on an IoT sensor node set are done by the proposed model, which is based on the ISFO model. In this article, the residual energy of the node, the center of the neighbor and the distance between the node and the sink are considered in which the set of CHs selected as $CH = CH_1, CH_2, \ldots, CH_s$ are defined. By selecting CH, the process of sending packets is possible with confidence. This is because the sensors communicate directly with CH instead of directly with base station, resulting in a reduction in energy when the packets are transmitted by CH to BS. Figure 2 indicates the flowchart of the proposed scheme. Algorithm 1 indicates the pseudocode of the Improved Sailfish Optimization (ISFO). In the pseudocode, the steps of the ISFO model are shown.

| Algorithm 1. Pseudocode of the improved Sailfish Optimization (ISFO) |
|---|
| Parameters initialization (A=4, ε= 0.001). |
| Calculate the fitness function of sailfish and sardines. |
| **While** the stop conditions are not satisfied |
|    **For** each sailfish |
|       Estimate $\lambda_i$ using Eq. (7). |
|       Update the location of sailfish using Eq. (15). |
|    **End For** |
| Calculate *AttackPower* using Eq. (10). |
|   **If** *AttackPower* < 0.5 |
|      Estimate α and β using Eq. (11) and Eq. (12). |
|      Choose a set of sardine base on the value of α and β |
|      Update the location of selected sardine by Eq. (18). |
|   **Else** |
|      Update the position of all sardine by the Eq. (18). |
|   **End If** |
|    Compute all sardine's fitness |
|   **If** there is a better solution |
|      Change a sailfish with injured Sardine using Eq. (13). |
|      Delete the hunted sardine |
|      Update the best sailfish and best sardine |
|   **End If** |
| **End While** |
| **Return** best sailfish |
| **//Function Clustering ()** |
| Node Deployment |
| Node Initialization |
| **While** (Round< $Max_{Rounds}$) |
| Clustering process using ISFO algorithm |
| Evaluation Fitness Function |
| Identify the best CH nodes with minimum fitness function |
| Keep the best solutions (CH) |
| Cluster member nodes transmit packet to the CH |
| CH nodes send packet to base station |
| Round termination condition |
| Return best vector with minimum fitness function |
| **End While** |
| **//Function SDN Controller ()** |
| **Start** |
| SDN Controller: Planned Clustering Implementation |
| BS: Send Request Massage from SDN Controller |
| SN: Revive Request Massage from BS |
| SN: Send Information (Such as position, Energy, and etc.) |
| BS: Receive Information (Such as position, Energy, and etc.) |
| CH: Clustering Operations |
| Command: Update Information |
| **End** |

## Improved SFO

The SFO algorithm uses two factors to search, so it can effectively improve population diversity and prevent population diversity shortcomings. Sailing populations of fish and sardines are randomly initialized. $s^{\min}$ and $d^{\min}$ represent the minimum number of fish and sardine sails, respectively. The value of both is equal to the minimum number of sensor

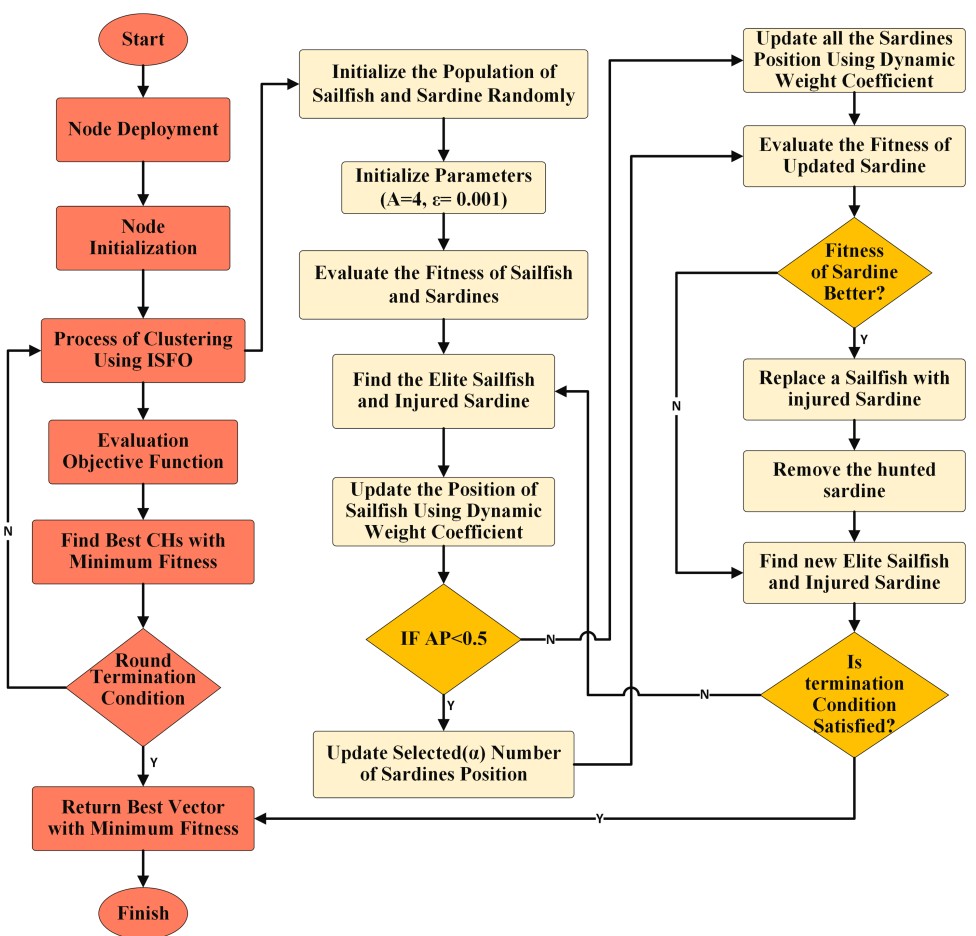

**Figure 2** The flowchart of the proposed scheme.

nodes. Similarly, $s^{max}$ and $d^{max}$ represent the maximum values of sail and sardines, both of which are equal to the maximum sensor nodes.

$$s_{i,j} = s^{min} + rand \times (s^{max} - s^{min})$$
$$d_{i,j} = d^{min} + rand \times (d^{max} - d^{min}) \tag{14}$$

The best solutions are regarded as vectors that CHs have a high residual energy. The dynamic weight coefficient $(w)$ is continuously entered in the formula for updating the position of the sailfish, which in the early stages is greater than the value of w and leads to global exploration. Finally, the value of w decreases comparatively, which is more useful for local search. Therefore, by Eq. (15) in iteration $i$, the new position of the sailfish ( $X^i_{new_{SF}}$ ) is updated. Weighting strategies are common in most swarm intelligence algorithms (*Ouyang, Qiu & Zhu, 2021*; *Wu et al., 2022*). In general, meta-heuristic algorithms partially reduce getting stuck in local optima by adaptively shifting between maximum and minimum values. Weighting in the initial stage weakens the effect of randomness and balances the

search mechanism in the problem space. Adaptive weighting improves the quality of agents' positions. It enables agents to converge to optimal positions faster and overall accelerates the rate of convergence.

$$X^i_{newSF} = \omega \times X^i_{elite_{SF}} - \lambda_i \times \left( rand(1,0) \times \left( \frac{X^i_{elite_{SF}} - X^i_{injured_S}}{2} \right) - X^i_{old_{SF}} \right) \tag{15}$$

where $X^i_{elite_{SF}}$ position of elite sail, $X^i_{injured_S}$ best position of damaged sardine, $X^i_{old_{SF}}$ current position of sail, and $\lambda_i$ is a coefficient in the $i$th iteration that is generated according to Eq. (16).

$$\lambda_i = 2 \times rand(0,1) \times PD - PD \tag{16}$$

PD indicates the number of baits per repetition. Since the number of baits decreases during group hunting by the sailfish, the PD parameter is an important parameter to improve the location of the sailfish around the bait. The formula of PD metric is defined according to Eq. (17).

$$PD = 1 - \left( \frac{N_{SF}}{N_{SF} + N_S} \right) \tag{17}$$

In the SFO algorithm, the new location of sardine $X^i_{new_S}$ is defined according to Eq. (18). By the parameter w, the search balance is created in the environment and the optimal solution is reached in the shortest possible time.

$$X^i_{new_S} = r \times \left( \omega \times X^i_{elite_{SF}} - X^i_{old_S} + AP \right) \tag{18}$$

The value of $\omega$ is defined by Eq. (19). In the Eq. (19) $i_{max}$ represents the maximum iteration.

$$\omega = \frac{e^{2 \times (1 - i/i_{max})} - e^{-2(1 - i/i_{max})}}{e^{2 \times (1 - i/i_{max})} + e^{-2(1 - i/i_{max})}} \tag{19}$$

Figure 3 shows the impact of w on the other factors. The weight factor (w) changed the position of $X_{injured}$ and $X_{elite}$. That is, the new position of the agents in the problem space is closer to the optimal solution. Like other population-based optimization algorithms, the SFO must strike a balance between exploration and exploitation. Fundamental SFO tends toward exploration, since the position-updated equation ignores the target point's location information and only utilizes it to determine the distance to the next searching zone at random. Additionally, the SFO's results for multi-modal issues show that its exploitation capability is lacking (*Zhang & Mo, 2022*).

The ISFO model uses elite sailfish in the current group to complete learning. The ISFO model has a different approach to updating sail and sardine that directly affects agent learning.

## Problem coding

In the ISFO, agents are considered as sensor nodes while their position indicates their performance. Agents can change the information about the multidimensional search space by their position. In ISFO with factor N, the position of the nth sensor in the iteration $t$ for

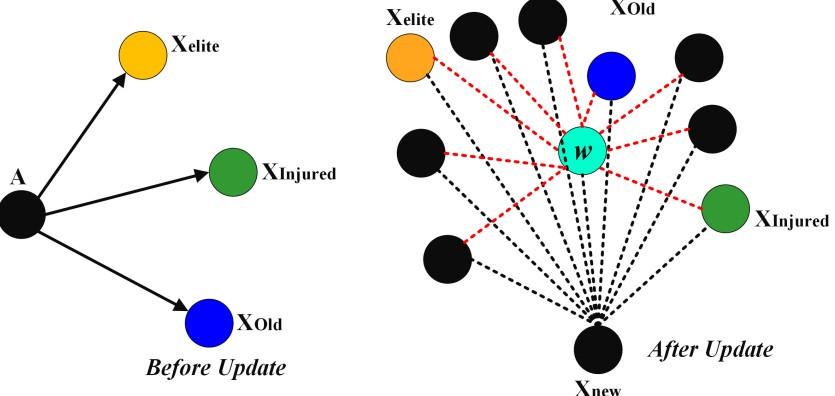

**Figure 3** The impact of w on the other factors.

$i = 1, 2, \ldots, N$ is defined according to Eq. (20), where $X_i^d(t)$ indicates the position of the i factor in the repetition of t in the dimension d and m refers to the dimension of the search space.

$$X_i(t) = \left(X_i^1(t), \ldots, X_i^d(t), \ldots, X_i^m(t)\right); i = 1, 2, \ldots, N \tag{20}$$

The sail of the fish moves in the search space to find its best local solution, and the quality of the solution produced by the sail is evaluated using the fit function. A change in the search behavior of sailfish leads to a change in the location of other sardines in the search space. To achieve the optimal answer, the sails of the fish in the search space change their position. The movement of each sailfish is influenced by the best sardines. As a result, the sailfish are guided to the best possible position (global solution). In the proposed model, each node has a chance to participate in the CH selection process, and the role of CH among all network nodes is to balance the consumption of energy. Selecting the thread in rotation mode among all nodes leads to uniform mode.

For example, if 100 sensor nodes are located in the IoT network, they are marked with node IDs from 1 to 100. Therefore, the value of each sailfish $X_{i,d}$ is determined as $1 \leq X_{i,d} \leq 100$. The proposed model starts with the process of validating the energy of the nodes, in which the nodes are considered as a set of factors. Figure 4 shows that in a 10-member vector, nodes 15, 23, 5, 58, 92, 71, 35, 11, 87, and 46 were selected as CH nodes. The selected CHs are the position of the sailfish. The position of the sailfish changes when it reaches the sardine. Every change is creating an optimal solution.

### Fitness function

The fitness function is considered as the main factor for selecting the CH nodes in the proposed model. Since the solutions of the agents depend on the fit function, the fitness function of the proposed model is defined based on the residual energy parameters, the distance between the clusters and the distance between the sensors and the sink. The rate of weights ($\omega_1$, $\omega_2$, $\omega_3$) is determined based on different tests. If the weights are not adjusted, the value of the fitness function moves towards the maximum.

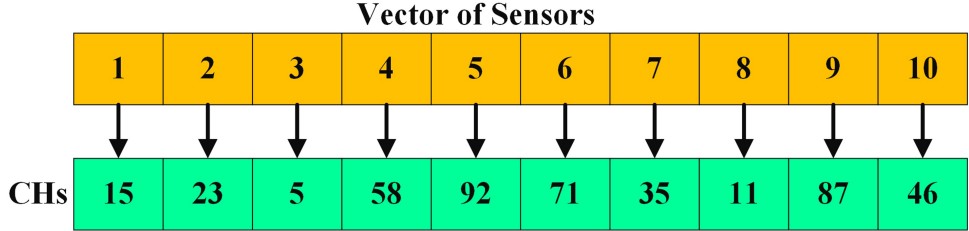

**Figure 4** In a 10-member vector, nodes 15, 23, 5, 58, 92, 71, 35, 11, 87, and 46 were selected as CH nodes.

$$minimize = \omega_1 \times f_1 + \omega_2 \times f_2 + \omega_3 \times f_3$$

$$where \ \omega_1 + \omega_2 + \omega_3 = 1 \ and \ 0 < \omega_1, \omega_2, \omega_3 < 1. \tag{21}$$

Residual energy (f1): The remaining energy prevents dead nodes from being selected as CH. CH involves collecting data from non-CH nodes and transferring the data to the sink after aggregation. The best headwaters are selected based on the residual energy and the distance to the neighboring nodes to determine the optimal paths to the sink node. Hence, the sensor node with maximum remaining energy (f1) is considered as CH. In Eq. (22), $E_{CH_i}$ represents the residual energy of $CH_i$, and m determines the number of threads.

$$f_1 = \sum_{i=1}^{m} \frac{1}{E_{CH_i}} \tag{22}$$

Intra-cluster distance (f2): Indicates the Euclidean distance between the members of the cluster and the CH nodes. If this distance is minimal, then the energy required to process the data is also reduced. Therefore, the goal of f2 is to achieve the minimum distance based on the distance within the cluster. $I_{m_j}$ Specifies the nodes within a cluster of j cluster.

$$f_2 = \sum_{j=1}^{m} \left( \sum_{i=1}^{I_{m_j}} d(s_i, CH_j) / I_{m_j} \right) \tag{23}$$

The distance between the sensor nodes and CH is determined using the distance matrix D (L × M) according to Eq. (24). The Euclidean distance between CH and a normal node is described as $CH_c$, and the sensor nodes are defined as $s_1, s_2, \ldots, s_n$. Two nodes, like a normal node, are denoted by $i$ and a thread by j, and their positions are denoted by u and v. Eq. (25) is used to calculate the Euclidean distance between the sensor node and the CH node.

$$D(L*M) = \begin{bmatrix} CH_{c_1,s_1} & CH_{c_1,s_2} & \cdots & CH_{c_1,s_n} \\ CH_{c_2,s_1} & CH_{c_2,s_2} & \cdots & CH_{c_2,s_n} \\ \vdots & \vdots & \vdots & \vdots \\ CH_{c_k,s_1} & CH_{c_k,s_2} & \cdots & CH_{c_k,s_k} \end{bmatrix} \tag{24}$$

$$d_{i,j} = \sqrt{(j_u - i_u)^2 + (j_v - i_v)^2} \tag{25}$$

In data transferring phase, a time slot is provided for each node. The main purpose of each node is to collect data and send it to the header. Once data has been collected from all nodes within the cluster, the header node transmits the associated data to the BS.

Distance between CH and sink (f3): Indicates Euclidean distance between CH and sink. The sensor node with more residual energy, less distance with neighboring nodes and the sink node is selected as the CH node using ISFO. Consequently, the strategy of the proposed model for selecting CH is the average distance between nodes to send packets to BS. As a result, the shortest distance to the sink is considered. SN indicates the sink node.

$$f_3 = \sum_{i=1}^{m} d(CH_j, SN) \tag{26}$$

## Terms of completion

The iterative phase is continued until the halting requirements are met or the algorithm has completed the maximum number of cycles. The population of sailfish is updated during this operation. The location of the entire sardine must be updated, or the attack power determines the position of the selected sardines. When there are no other assailants present, the sailfish alter their position against the victim. Other parameter values have been analyzed, and the ISFO algorithm determines the attack alteration method throughout hunting. The fitness value of all sardines is evaluated and sardine in solution is replaced. Finally, the best sailfish and sardine have been upgraded. The entire process is repeated till the maximum number of iterations is reached.

## Computational complexity

The increase in computational complexity in meta-heuristic algorithms is dependent on the number of iterations, changes in the initial population, and changes in updating the agents' positions. The increase in the number of repetitions and population diversity in the proposed model is insignificant. In the SFO algorithm, due to the proper balance between exploration and exploitation, the increase in computational complexity is low. Compared with the standard SFO, the proposed model increases the complexity of executing the new position of the sailfish and computing individual position. Other operators do not add complexity.

## Model of consuming energy

The energy consumption model is evaluated by the radio energy dissipation model according to the distance between the receiver and the transmitter. The effectiveness of clustering models is frequently assessed using the metric of energy usage. The entire of energy required by sensor nodes during network operation is known as energy consumption. The distance between the source and the destination, the retransmission rate, and the control messages all have an impact on this measure. This ratio should be

**Table 1  Shows the important simulation parameters.**

| Parameter | Value |
|---|---|
| Number of sensors | 150, 300 |
| Area | 200 m ×200 m |
| E | 0.5 J |
| $E_{elec}$ | 50nJ/bit |
| $E_{mp}$ | 0.001310pJ/bit/m$^4$ |
| $E_{fs}$ | 10pJ/bit/m$^2$ |
| Data packet size/B | 2048 |
| Number of rounds | 1700 |
| Type of nodes | Static |
| Population size | 30 |
| Maximum iterations for ISFO | 150 |
| A | 4 |
| $\varepsilon$ | 0.001 |

kept to a minimum to reflect efficient energy use. The energy allocated to send an L-bit packet at distance d is defined according to Eq. (27).

$$E_{TX}(L, D) = \begin{cases} L.E_{elec} + L.E_{fs} \times d^2 & (d < d_0) \\ L.E_{elec} + L.E_{mp} \times d^2 & (d \geq d_0) \end{cases} \tag{27}$$

$$E_{RX}(l) = l \times E_{elec} \tag{28}$$

$$d_0 = \sqrt{E_{fs}/E_{mp}} \tag{29}$$

When a sensor node receives l-bit packet, its energy consumption is calculated according to Eq. (28). where $E_{elec}$, $E_{mp}$ and $E_{fs}$ are constant parameters. $E_{TX}(L, d)$, $E_{RX}(l)$ and d representing the energy consumed to send and receive a 1-bit data packet and, the distance between the transmitter and receiver respectively. $d_0$ is the threshold value. The $E_{fs}$ parameter expresses the energy consumption of the transmission amplifier for free routing and cooperation between nodes. The $E_{mp}$ parameter expresses the power consumption of the transmission amplifier for multi-route routing.

## EVALUATION AND RESULTS

This article has been evaluated in Python and IoT libraries. The evaluation of the ISFO model is based on 150 and 300 nodes that are distributed in an area of 200 m × 200 m based on random. Table 1 shows the important simulation parameters. All results are reported based on the average of 20 runs. The ISFO model is compared with the main clustering models, namely LEACH, LEACH-E (*Xu et al., 2012*). The SFO is also used for comparison to show the ISFO model performance. The number of iterations and the initial population in the algorithms are equal. Table 1 shows the simulation parameters.

### Number of alive nodes
Figure 5 shows the number of alive nodes for the 150 and 300 nodes. The number of alive nodes for the ISFO model in 500 and 1,000 rounds with 150 nodes included 138 and 94

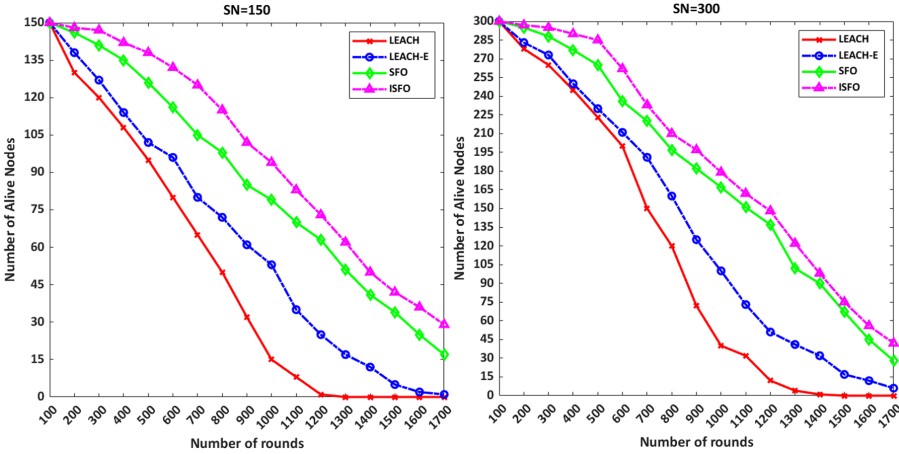

**Figure 5** **The number of alive nodes for the 150 and 300 nodes.**

nodes. For 300 sensor nodes, the number of alive nodes for ISFO and SFO per 1,000 rounds is 179 and 167, respectively. For 300 sensor nodes, the number of alive nodes for ISFO and SFO per 1,000 rounds is 179 and 167, respectively. In the ISFO model, the alive nodes are still stable after the simulation round is completed because the ISFO model has achieved the best degree of convergence by improving the solutions, and the agents in the environment have chosen the best cluster. From the results of the diagrams, it is noteworthy that the ISFO model has a good performance compared to the LEACH and LEACH-E in term of network lifetime.

In Fig. 6 the results are shown for 2,500 rounds. The chart results show that the ISFO model has more live nodes compared to other models. The purpose of 2,500 rounds are to show the convergence chart of the models. The convergence graph of the number of live nodes with 1,700 rounds is not clear. But the comparison between 1,700 rounds and 2,500 rounds shows that the convergence of dead nodes in the ISFO model is done in a smoother form. The number of live nodes after 2,500 rounds for 150 and 300 nodes mode by ISFO model is two nodes and five nodes, respectively. The number of live nodes after 2,500 rounds for 150 and 300 nodes mode by SFO model is equal to 0.

Table 2 shows the number of clusters for each model based on different runs. The proposed model decreases the number of cluster when compared to LEACH, LEACH-E, SFO, and ISFO. According to the Table 2, it is clear that the number of clusters is different in each run. Therefore, the results will be different based on the number of clusters. If the number of clusters is less, then the energy consumption between the cluster heads will be less. Therefore, the lifetime of the network increases. A large number of clusters leads to network interference and lost packets. In the ISFO model, the number of clusters is optimal.

## Number of delivered packets

Figure 7 shows the number of packets delivered to the sink for the ISFO model and the LEACH, LEACH-E and SFO models. The number of received packets in 1,000 rounds with

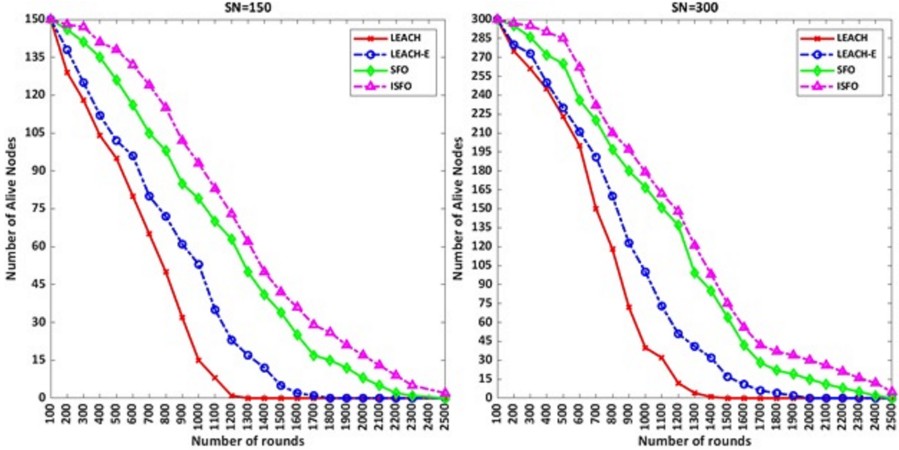

**Figure 6** The number of alive nodes for 2,500 rounds.

**Table 2** Number of clusters for each model based on different executions.

| Runs | Nodes | LEACH | LEACH-E | SFO | ISFO |
|------|-------|-------|---------|-----|------|
| 1 | 150 | 22 | 20 | 18 | 15 |
| | 300 | 25 | 22 | 20 | 18 |
| 2 | 150 | 23 | 21 | 19 | 14 |
| | 300 | 25 | 22 | 20 | 19 |
| 3 | 150 | 22 | 20 | 18 | 13 |
| | 300 | 23 | 22 | 23 | 17 |
| 4 | 150 | 22 | 20 | 18 | 16 |
| | 300 | 25 | 24 | 20 | 18 |
| 5 | 150 | 21 | 20 | 21 | 15 |
| | 300 | 24 | 23 | 21 | 16 |
| 6 | 150 | 22 | 20 | 18 | 14 |
| | 300 | 26 | 26 | 20 | 18 |
| 7 | 150 | 24 | 20 | 18 | 13 |
| | 300 | 25 | 22 | 20 | 17 |
| 8 | 150 | 22 | 21 | 17 | 15 |
| | 300 | 25 | 25 | 23 | 18 |
| 9 | 150 | 23 | 20 | 18 | 16 |
| | 300 | 25 | 22 | 20 | 17 |
| 10 | 150 | 25 | 22 | 20 | 14 |
| | 300 | 27 | 24 | 21 | 18 |

150 nodes by ISFO and SFO models is equal to 2,856 and 3,523, respectively. The number of received packets in 1,500 rounds with 150 nodes by ISFO and SFO models is 5,326 and 4,156, respectively. The number of received packets in 500 rounds with 300 nodes by ISFO and SFO models is equal to 2,059 and 1,847, respectively. The number of received packets in 1,200 rounds with 300 nodes in ISFO and LEACH-E models is equal to 5,287 and 3,055,

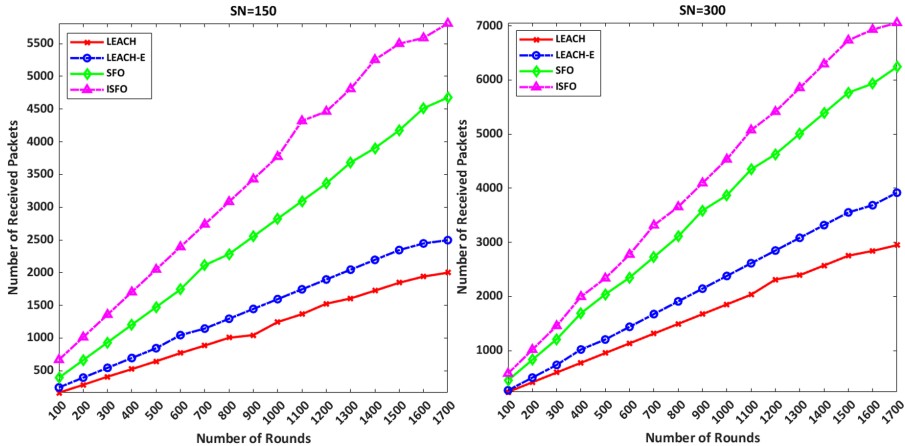

**Figure 7 The number of packets delivered to the sink for the ISFO model and the LEACH, LEACH-E and SFO models.**

respectively. The number of received packets in 1,200 rounds with 150 nodes in LEACH and LEACH-E models is equal to 1,502 and 1,594, respectively. Comparisons show that the ISFO model has been able to increase the number of packages due to the selection of optimal clusters.

## Packet delivery rate

Figure 8 shows a comparison of package delivery rates for the ISFO model and the LEACH, LEACH-E and SFO models. Package delivery rates are defined according to Eq. (30) (*Guleria et al., 2021*).

$$PDR = \frac{\text{Number of Packets Received}}{\text{Number of Packets Transmitted}} \times 100 \tag{30}$$

If PDR is too low, these models will not be able to send the packets to the sink completely. In Fig. 8, the X-Axis indicates the simulation rounds and the Y axis indicates the packet delivery rate (%). Figure 8 shows that the PDR of the proposed model is higher than other models.

Table 3 shows the number of nodes with respect to PDR. The ISFO increases the PDR when compared to LEACH, LEACH-E, and SFO. It is observed that the PDR increases as the number of nodes increases. It is mainly due to consideration of ISFO. For 150 nodes, the PDR of the ISFO model is 89.44%. PDR of the ISFO model is 94.47% on 300 nodes. PDR of the SFO is 87.40% on 300 nodes.

## Throughput

Transmitting the volume of data packets throughout the simulation period is considered as throughput. Operating power is defined according to Eq. (31).

$$Throughput = \frac{\text{number of data packets sent (bits)}}{\text{Time period (seconds)}}. \tag{31}$$

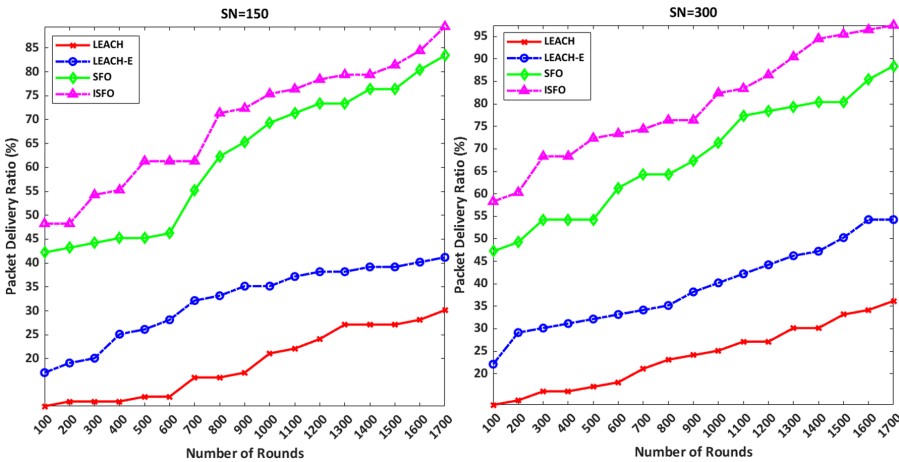

**Figure 8** A comparison of package delivery rates for the ISFO model and the LEACH, LEACH-E and SFO models.

**Table 3** Number of nodes vs. PDR.

| Number of Nodes | PDR (%) | | | |
|---|---|---|---|---|
| | LEACH | LEACH-E | SFO | ISFO |
| 150 | 32.16 | 46.23 | 83.41 | 89.44 |
| 200 | 33.87 | 47.64 | 84.85 | 91.53 |
| 250 | 35.49 | 49.25 | 85.93 | 92.48 |
| 300 | 36.18 | 53.26 | 87.40 | 94.47 |

In Fig. 9, the X and Y axes represent the number of rounds and throughput in bits per second, respectively. The ISFO model provides $3.5 \times 10^4$ bit/s throughput compared to other models up to the end of 1700 rounds with 150 nodes. LEACH, LEACH-E models achieved $2.2 \times 10^4$, $2.4 \times 10^4$ bps in the last simulation round with 150 nodes. LEACH, LEACH-E models achieved $2.1 \times 10^4$, $2.3 \times 10^4$ bps in the last simulation round with 300 nodes, respectively. According to the results, the ISFO model has more throughput than the LEACH, LEACH-E models.

## Remaining energy

Figure 10 shows the average residual energy for the 150 and 300 sensor nodes. It can be seen that at 1,700 rounds with 150 and 300 nodes, the residual energy of the ISFO scheme is higher than other schemes. LEACH, LEACH-E models are weaker than ISFO and SFO. Because traditional models do not benefit from the function of fit and selection of optimal solutions. The average residual energy in 900 rounds with 150 nodes by ISFO and SFO models is 51 and 47 joules, respectively. The average residual energy in 900 rounds with 150 nodes by ISFO and LEACH models is 51 and 23 joules, respectively. The average residual energy at 1,300 rounds with 150 nodes by LEACH-E and SFO is 19 and 31 joules, respectively. The average residual energy at 1,700 rounds with 300 nodes by the ISFO and

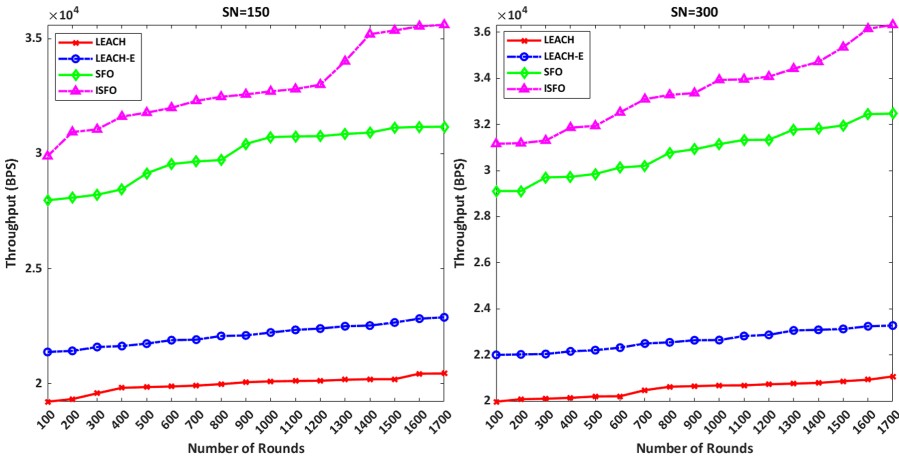

**Figure 9** The X and Y axes represent the number of rounds and throughput in bits per second, respectively.

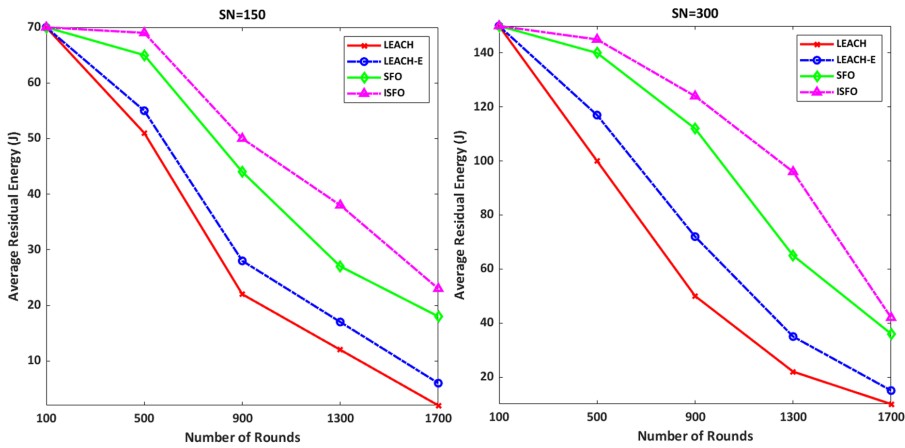

**Figure 10** The average residual energy for the 150 and 300 sensor nodes.

SFO models is 42 and 37 joules, respectively. The average residual energy in 500 rounds with 300 nodes by ISFO and LEACH-E is 147 and 119 joules, respectively.

The statistical analysis of the proposed model and other models based on residual energy is shown in Table 4. This may prove the efficiency of the ISFO model over the other models in terms of remaining energy. The metrics of minimum value, mean, maximum value and standard deviation (SD) are specified and computed for the models based on the results of the remaining energy of number of nodes. It was found that the LEACH, LEACH-E, and SFO models all performed significantly worse than the ISFO.

In Table 5, the statistical significance of the residual energy for one round is shown using the paired t test. In all cases, $P < 0.05$, so the null hypothesis is rejected at the 5% significance level and the alternative hypothesis is accepted at the 95% confidence level.

**Table 4  Statistical analysis of the proposed model and other models based on residual energy.**

| Number of Nodes | Metrics | LEACH | LEACH-E | SFO | ISFO |
|---|---|---|---|---|---|
|  | Minimum | 2 | 6 | 18 | 23 |
| 150 | Mean | 31.40 | 35.20 | 44.80 | 50 |
|  | SD | 28.29 | 26.64 | 22.79 | 20.21 |
|  | Minimum | 10 | 15 | 36 | 42 |
| 300 | Mean | 66.40 | 77.80 | 100.60 | 111.40 |
|  | SD | 58.17 | 56.03 | 48.89 | 44.24 |

**Table 5  Results of $t$-test of residual energy for a single round.**

| ISFO | $t$-Test | Significance of the null hypothesis | Confidence Interval 95% | |
|---|---|---|---|---|
|  |  |  | Lower | Upper |
| LEACH | 3.26 | <5% | 3.21 | 3.52 |
| LEACH-E | 4.35 | <5% | 4.28 | 4.68 |
| SFO | 7.64 | <5% | 7.61 | 8.21 |

CH selection in the proposed model reduces energy consumption based on the distance between clusters.

In addition, the significance of the proposed model's performance was evaluated using statistical tests, and it was found to be significant with a confidence level of 95%. During this procedure of testing, the samples connected to the amount of leftover energy were chosen for each algorithm that was being investigated. This was done in order to ensure accurate results. Figures 11 and 12 highlights the descriptive investigation of the proposed model and benchmarked algorithms with respect to residual energy. It can be seen that the ISFO showed the best performance for 150 sensors, with median value of 50. Also, it can be seen that the ISFO showed the best performance for 300 sensors, with median value of 124.

## CONCLUSION AND FUTURE WORKS

The goal of the IoT is to simplify operations and processes for intelligent tasks with minimum human intervention using various processing systems. IoT applications run based on a set of policies, algorithms, and principles that are programmed into the IoT framework. In this article, a clustering protocol called ISFO is proposed to achieve energy efficient use in SDN-IoT network. The purpose of the SDN was to program the IoT environment using the ISFO model. In the ISFO model, suitable CHs were selected based on the energy threshold and the distance between CHs and member nodes. Comparison between the proposed method and other models showed that the proposed model had a longer network life than LEACH, LEACH-E. The ISFO model was used to improve the solution vectors and select the best cluster. The ISFO model was tested with two scenarios including 150 and 300 sensor nodes based on different factors. The results showed that the ISFO model obtained more live nodes compared to other models and SFO. Also, the

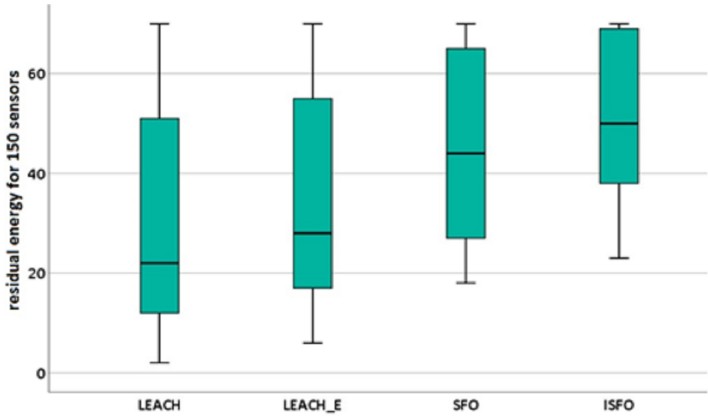

**Figure 11  Box plot for residual energy (150 sensors).** The descriptive investigation of the proposed model and benchmarked algorithms with respect to residual energy.

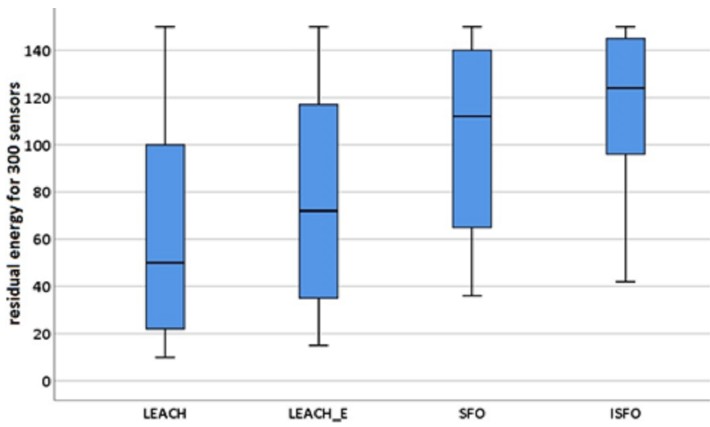

**Figure 12  Box plot for residual energy (300 sensors).** The descriptive investigation of the proposed model and benchmarked algorithms with respect to residual energy.

delivery rate of packages in ISFO model was higher compared to LEACH and LEACH-E. In general, the results of ISFO model compared to other models on the average residual energy factor with 150 and 300 nodes have improved by 23.41% and 28.79%, respectively. For future work, the SDN-IoT performance will be tested using optimization algorithms and a combination of fuzzy logic methods to find the optimal CH nodes.

### Funding
The authors received no funding for this work.

## Competing Interests

Sedat Akleylek is an Academic Editor and a Section Editor of Cryptography, Security and Privacy for PeerJ.

## Author Contributions

- Ramin Mohammadi conceived and designed the experiments, performed the experiments, analyzed the data, performed the computation work, prepared figures and/or tables, authored or reviewed drafts of the article, and approved the final draft.
- Sedat Akleylek conceived and designed the experiments, analyzed the data, performed the computation work, authored or reviewed drafts of the article, and approved the final draft.
- Ali Ghaffari conceived and designed the experiments, performed the experiments, performed the computation work, prepared figures and/or tables, authored or reviewed drafts of the article, and approved the final draft.

## Data Availability

The code is available in the Supplemental File.

## Supplemental Information

Supplemental information for this article can be found online at http://dx.doi.org/10.7717/peerj-cs.1424#supplemental-information.

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
