# Peer review of "SDN-IoT: SDN-based efficient clustering scheme for IoT using improved Sailfish optimization algorithm"

_PeerJ Computer Science, doi:10.7717/peerj-cs.1424_

## Round 0.1 · original submission · Major Revisions

From the reviewers' comments, the authors are suggested to revise the paper. Please remember to highlight the modified parts together with a point-by-point response letter.

Reviewer 1 ·

Basic reporting

1. At line 287, no pseudocode is given for the Improved Sailfish Optimization (ISFO).
2. At line 353 and 355,what is a well node? Is it sink?

Experimental design

no comment

Validity of the findings

no comment

Reviewer 2 ·

Basic reporting

1. In lines 416-417, the number of received packets in 1000 rounds with 150 nodes by SFO models is equal to 3215. However, as can be seen in Figure 6, it clearly does not receive more than 3000 packets.

2. In lines 451-452, the average residual energy at 1300 rounds with 150 nodes by LEACH-E and SFO is 20 and 35 Joules, respectively. However, in Figure 9, the average residual energy of LEACH-E did not reach 20 J and the average residual energy of SFO did not exceed 30 J

Experimental design

no comment

Validity of the findings

no comment

Reviewer 3 ·

Basic reporting

The paper proposes an improved meta heuristic for clustering of nodes in IoT environments, aimed at minimizing energy consumption. he manuscript is correctly organized, and the problem is well motivated too. The background material is sufficient and the technical depth of the paper is appropriate. References are up to date, most of them very recent, but self-citations are frequent and should be reconsidered (5 papers contain self-citations).
Both the quality of the figures and the English writing can be improved to enhance clarity. The mathematical notation is rather unclean, with many symbols having multiple subscripts and superscripts (e.g., the positions of sailfish and sardines), which tend to hinder the reading.

Experimental design

The key contribution is a modification to the metaheuristic (the sailfish algorithm) that eventually yields an improvement in achieving good clustering in a population. This is exploited to optimize the positions of the cluster heads in a dense network of IoT devices, thus achieving better performance and less energy consumption (so a longer lifetime).

The problem is well formulated and exposed with clarity. The original SFO algorithm is presented in detail, but not with much explanation on the interpretation of the equations (6)-(13). though it is not difficult to follow it. Technically, there are a number of issues that would require better explanation:

1) There is not a clear explanation or proof that the modified equations (15) and (18) containing the new factor w actually produce a new position that is 'closer to the optimal solution' (lines 320-321) as the authors claim. Please, substantiate this claim. Figure 3 is not a proper explanation.

2) Though it is implicit, it is not stated in the paper that the chosen cluster heads are the positions of the sail fish. Is this correct?

3) The fitness function merges three functions, two of which are related to the distance in the cluster and one with energy (residual). Relative weights are used for each part, but the authors do not explain how to set these or the numerical value used in the simulations. Provide the details.

4) Some important information for the simulation is missing. How are the nodes distributed? Do their locations change in different simulation runs? How many simulated were executed for the results (ISFO is stochastic)?

5) The paper is titled "SDN-based efficient clustering..." However, SDN plays no role in the proposed algorithm, and has actually nothing to do with the proposed solution. This is a misleading title, as is part of the Introduction dealing with the principles of SDN.

Validity of the findings

There is a significant improvement in performance over the original clustering algorithm, but the experimental design is rather limited to generalize this idea.

1) The authors should present in full detail the simulation environment used to get their results: all the parameters, the stochastic geometry model (if used) and a much netter explanation if the energy consumption model (for instance, some terms in (27), (28) and (29) are not defined). Similarly, it is not clearly stated how many clusters are formed, or how its number is determined.

2) The contribution of the paper is incremental (a modification to a known algorithm), so provide enough experimental support that the algorithm improves substantially its original version.

Additional comments

A thorough revision of English is necessary along the paper,

Reviewer 4 ·

Basic reporting

The article presents a heuristic method for SDN-based clustering in IoT.
The related works section could be extended by strengthening the comparison among the related works and between the related works and the present article, to improve the motivation about using the Sailfish optimization to tackle the identified problem.
The article should open source the code and the raw data for reproducibility.

Experimental design

There is no statistical analysis performed (e.g., confidence intervals, repetitions, etc.)
The decision of evaluating the method compared with the baselines only over rounds is unclear. More parameters should be studied to check how the proposed method performs in a wider range of scenarios.

Validity of the findings

Due to the closed-source nature of code and data, and the reduced details on the simulation setups, it is difficult to reproduce the experiment, limiting the validity of the findings.

---

## Round 0.2 · Major Revisions

From the reviewers' comments, the authors are suggested to revise the paper again. Please remember to highlight the modified parts together with a point-by-point response letter.

Reviewer 3 ·

Basic reporting

No comment.

Experimental design

In the revised version of the manuscript, the experimental design has been written with further and sufficient detail. The described methodology is correct, and the description allows for a replicable experiment by other researchers. This Section has been significantly improved over the previous manuscript.

Validity of the findings

The contribution is essentially a modification of the heuristic Sailfish algorithm, by introducing in it a new random parameter that improves exploration. The contribution is incremental and adds complexity to the heuristic approach for optimization (Sailfish already has a high number gf tunable parameters). However, the performance results are good and may justify the increased complexity.

Heuristic algorithms lack a solid theoretical foundation and are subject to many environmental factors, so they can be tuned to match almost any objective functions just by trial and error. Leaving this point apart, the paper makes an incremental contribution in this area by introducing another randomized factor into the algorithm.

Additional comments

The revised version contains more details, more clarity and a better explanation of the results. The contribution is minor but valuable. The paper can be published in its current form.

Reviewer 4 ·

Basic reporting

The authors have improved the article's text and have shared their source code to strengthen verifiability. Adding an online reference to the source code in the article would improve reproducibility for other researchers, but it is not a journal requirement. A few comments from the previous review round are, however, left untackled, especially from the standpoint of experimental evaluation.

Experimental design

The measures of performance over the number of rounds should be converted in aggregated measures, in which a single numerical metric is produced per experiment instead of an evolution over time. Showing a trend over rounds (as done in the current plots) does not help the reader understand the general performance of the proposed method in generic scenarios or when the system's features change or to how the scenario's parameters impact the proposed method's performance. In some cases, the diagrams are truncated before the curves converge.

Validity of the findings

The authors have added standard deviation and minimum as statistical analyses for the results in Table 4 (which does not include a measurement unit). However, a major reviewer's comment is left untackled, i.e., adding confidence intervals to the plots to confirm statistical significance.

---

## Round 0.3 · Major Revisions

The authors still have not addressed the comments from the reviewer. If the authors can not fix the problems and improve the paper, the paper will be ejected without another chance to revise.

Please remember to highlight the modified parts with color and together with a point-by-point response letter.

Reviewer 4 ·

Basic reporting

-

Experimental design

Some of the issues previously raised by the reviewer are left untackled. Namely, the plots do not report confidence intervals, and the text does not indicate the number of repetitions of the simulation experiment.

Validity of the findings

The lack of confidence intervals on the plots makes it impossible for the reader to understand the strength of the displayed results.

Additional comments

-

---

## Round 0.4 · accepted · Accept

The authors have finished the revisions. No more comments.